# RECKONING: Reasoning through Dynamic Knowledge Encoding

**Zeming Chen**[1]    **Gail Weiss**[1]    **Eric Mitchell**[2]    **Asli Celikyilmaz**[3]    **Antoine Bosselut**[1]

EPFL[1]    Stanford University[2]    Meta AI Research[3]

{zeming.chen, gail.weiss, antoine.bosselut}@epfl.ch
eric.mitchell@cs.stanford.edu   aslic@meta.com

## Abstract

Recent studies on transformer-based language models show that they can answer questions by reasoning over knowledge provided as part of the context (i.e., in-context reasoning). However, since the available knowledge is often not filtered for a particular question, in-context reasoning can be sensitive to distractor facts, additional content that is irrelevant to a question but that may be relevant for a different question (i.e., not necessarily random noise). In these situations, the model fails to distinguish the necessary knowledge to answer the question, leading to spurious reasoning and degraded performance. This reasoning failure contrasts with the model's apparent ability to distinguish its contextual knowledge from all the knowledge it has memorized during pre-training. Following this observation, we propose teaching the model to reason more robustly by folding the provided contextual knowledge into the model's parameters before presenting it with a question. Our method, RECKONING, is a bi-level learning algorithm that teaches language models to reason by updating their parametric knowledge through back-propagation, allowing them to answer questions using the updated parameters. During training, the inner loop rapidly adapts a copy of the model weights to encode contextual knowledge into its parameters. In the outer loop, the model learns to use the updated weights to reproduce and answer reasoning questions about the memorized knowledge. Our experiments on three diverse multi-hop reasoning datasets show that RECKONING's performance improves over the in-context reasoning baseline (by up to 4.5%). We also find that compared to in-context reasoning, RECKONING generalizes better to longer reasoning chains unseen during training, is more robust to distractors in the context, and is computationally more efficient when multiple questions are asked about the same knowledge.

## 1   Introduction

Consider the sentence: "John is David's dad, and Tom is John's dad". Concluding that Tom is David's grandfather involves *reasoning* about the information in the sentence. Specifically, it requires understanding the direct information, or *contextual knowledge*, given in the sentence: the stated relationships between John, David, and Tom; and combining it with our existing, *commonsense knowledge* of the world: someone's dad's dad is their grandfather. Achieving such logical reasoning automatically has long been a goal of AI [17, 53, 74, 82].

The example above demonstrates two necessary abilities required for successful reasoning: first, holding large amounts of commonsense or general knowledge about the world, and second, processing and combining new information with existing knowledge. Transformer-based large language models have shown a remarkable capacity for the first of these abilities, repeatedly being demonstrated to memorize large amounts of data, or *parametric knowledge*, in their weights [8, 11, 50, 64].

37th Conference on Neural Information Processing Systems (NeurIPS 2023).

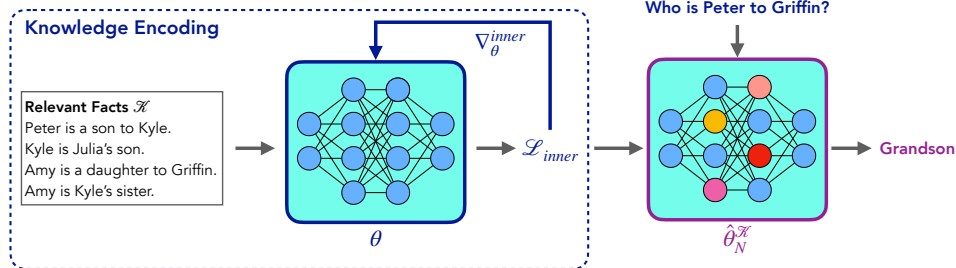

Figure 1: Our algorithm, RECKONING, solves reasoning problems by encoding external contextual knowledge into a model's parameters through gradient updates. At inference time, RECKONING performs a few parameter updates using the gradients of a language modeling loss to encode the relevant facts. Then, the updated model answers the question using only its implicit knowledge.

For the second, recent work showed that transformers fine-tuned to predict answers over a concatenated context ("The cow is big; If something is big then it chases the dog; If the cow chases the dog then the cow sees the rabbit") and question ("Did the cow see the rabbit?") achieve high performance on reasoning tasks where all necessary knowledge is given in the context [17]. We refer to this general setting as *in-context reasoning* (ICR).

In real-world question-answering settings [16, 22, 40, 42], large amounts of contextual knowledge may be provided at once, and the information may not be perfectly filtered for a specific question. Unfortunately, in-context reasoning is highly sensitive to *distractors* [70]: additional facts that are not relevant to a question (e.g., "The cow is round" for the above example). Indeed, when fine-tuning and evaluating GPT-2 [60] for ICR, we find that adding distractors to the context drops performance from $99.4\%$ to only $70.9\%$ accuracy for the same questions (§4.2). This sensitivity to distractors in contextual knowledge contrasts with GPT-2's apparent robustness to distractors in parametric knowledge: for any specific example, most of the training data seen by GPT-2—which forms its parameters—is likely to be completely irrelevant to that example. Naturally, we wonder whether presenting contextual knowledge in the same way as memorized knowledge, by encoding it into a model's parameters, will improve the reasoning abilities of transformer-based language models.

In this work, we propose a novel bi-level optimization algorithm, RECKONING, that learns to memorize (and reason) over facts (i.e., knowledge) by performing inference-time parameter updates using gradients computed from a language modeling loss on those facts. The updated model is then used to answer any questions about those facts. Our training framework involves two nested loops: the inner loop performs fast adaptations from a set of initial weights to memorize a set of external knowledge through a few gradient updates, and the outer loop optimizes those same initial weights such that the updated model will solve reasoning problems associated with the memorized knowledge. In other words, the outer loop learns optimal *meta-parameters* that can rapidly memorize and successfully reason over contextual knowledge, allowing knowledge memorization to be optimized directly for downstream reasoning. At inference time, instead of including external knowledge in the input sequence as the prefix to a question prompt, the model can encode it in its parameters through gradient updates and then reason over its updated parametric knowledge to reach a conclusion.

We evaluate RECKONING on two synthetic multi-hop reasoning datasets: ProofWriter [74] and CLUTRR-Systematic-Generalization (CLUTRR-SG) [29], and one real-world dataset, FOLIO [30], comparing against a fine-tuned ICR (FT-ICR) baseline that uses the same underlying model. Our results show that RECKONING consistently outperforms the FT-ICR baseline on each benchmark, demonstrating that it successfully learns to answer multi-hop reasoning questions as desired. In particular, we find that RECKONING more successfully generalizes to adversarial settings, such as the presence of distractor facts and the introduction of longer reasoning chains at inference time. Finally, while the inference-time gradient updates make RECKONING slower to process new knowledge than a typical ICR forward pass, our run-time analysis shows that RECKONING is more efficient when answering multiple questions about a shared knowledge set. This speedup occurs because RECKONING only needs to encode the knowledge once to answer multiple questions about it. Overall, we demonstrate that RECKONING is an effective algorithm for reasoning through dynamic and controllable knowledge encoding, overcoming an observed weakness in the common reasoning setting and providing multiple additional benefits.

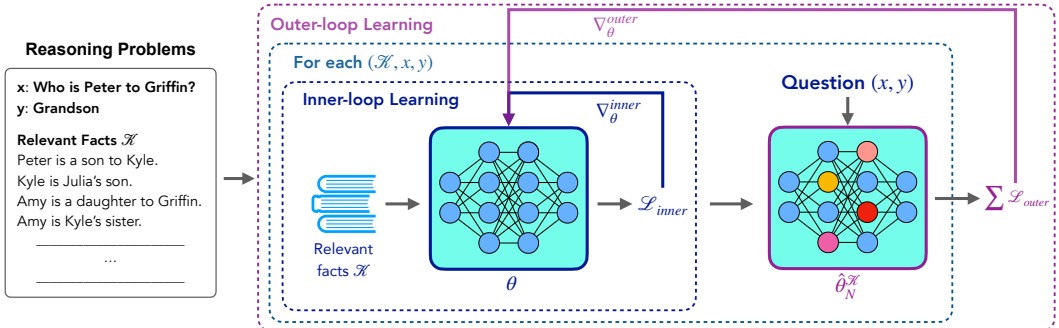

Figure 2: The two-stage training process of RECKONING with an inner and outer loop.

## 2 Background

**Notation** We use $f : \mathcal{X} \times \theta \to \mathcal{Y}$ to refer to parameterised functions in which $\mathcal{X}$ is the set of possible inputs and $\theta$ are their possible weights (parameters). We use $f_{\boldsymbol{\theta}} : x \mapsto f(x, \boldsymbol{\theta})$ to easily refer to any $f$ with a given set of parameters $\boldsymbol{\theta}$. We describe reasoning problems using tuples $(\mathcal{K}, \boldsymbol{x}, y^*, Y)$ such that $y \in Y$ is the correct answer for the question $\boldsymbol{x}$ given facts $\mathcal{K}$, and use $\mathcal{D}$ to refer to sets of such problems. When it is clear from context, we drop $Y$ and use only $(\mathcal{K}, \boldsymbol{x}, y^*)$.

**Language Modeling and Memorization** In the causal language modeling (CLM) objective, a parameterized model $f_\theta$ is trained to estimate the conditional probabilities of each token in a sequence given its predecessors: $p(x_t|x_{<t})$. Specifically, we train $f_{\boldsymbol{\theta}}$ to approximate $p$ using the CLM loss:

$$\mathcal{L}_{\text{CLM}}(f_{\boldsymbol{\theta}}, \boldsymbol{x}) = -\sum_{t=1}^{T} \log f_{\boldsymbol{\theta}}(x_t|x_1, ..., x_{t-1}). \tag{1}$$

This training objective allows language models to *memorize* individual training examples [10, 11], and we will exploit this ability to memorize and draw on contextual knowledge in our work.

**Transformers as Soft Reasoners** In natural language *reasoning* tasks, we are given reasoning problems $(\mathcal{K}, \boldsymbol{x}, y^*, Y)$ in natural language and attempt to recover the correct answer $y^*$ from the context $\mathcal{K}$, question $\boldsymbol{x}$, and possible answers $Y$ alone. In *in-context reasoning*, language models $f_{\boldsymbol{\theta}}$ trained with a CLM objective are applied to this task by selecting as the response the answer $y \in Y$ with a maximum probability according to the model's next-token prediction from the concatenated context and question: $y = \arg \max_{y' \in Y} f_{\boldsymbol{\theta}}(y'|[\mathcal{K}; \boldsymbol{x}])$. Previous works show that, after relevant supervised fine-tuning, transformer language models can achieve high performance in this setting [17, 29, 74], though this degrades significantly in the presence of irrelevant facts (*distractors*) [70].

## 3 Method

Addressing these challenges, we propose RECKONING (**RE**asoning through dynami**C K**n**O**wledge e**N**cod**ING**), which solves reasoning problems by memorizing the provided contextual knowledge, and then using this encoded knowledge when prompted with downstream questions. Specifically, RECKONING uses bi-level optimization to learn a set of meta-parameters primed to encode relevant knowledge in a limited number of gradient steps. The model can then use its updated weights to solve reasoning problems over this knowledge, *without further presentation of the knowledge itself.*

**Overview: Inference** Given a reasoning problem $(\mathcal{K}, \boldsymbol{x}, y, Y)$, we initialize our model with weights copied from a set of meta-parameters $\boldsymbol{\theta}$ and perform a constant number $N$ of gradient descent steps on these with the goal of minimizing the CLM objective on the knowledge set $\mathcal{K}$. This allows the model to memorize $\mathcal{K}$ in its updated parameters, which we refer to as $\hat{\boldsymbol{\theta}}_N^{\mathcal{K}}$. Next, we pass the question $\boldsymbol{x}$ to the model, using $f_{\hat{\boldsymbol{\theta}}_N^{\mathcal{K}}}$ to obtain a distribution over $Y$, and taking as output the answer $y \in Y$ with the highest probability. For this method to consistently output the ground truth $y^*$, we seek a set of optimal meta-parameters $\boldsymbol{\theta}^*$ that can quickly memorize (i.e., learn) the given knowledge in a way that then allows accurate reasoning when queried about the knowledge downstream.

**Training RECKONING**    Similar to the meta-learning algorithm MAML [25], which focuses on optimizing the initial parameters of a model so that the model can be good at learning new tasks with a few examples, we also optimize the initial parameters such that the model learns to encode knowledge for more effective reasoning. Given a distribution $p(\mathcal{D})$ of reasoning problems, our proposed bi-level optimization framework RECKONING (seen in Figure 2) optimizes the following objective:

$$\boldsymbol{\theta}^* \in \arg\min_{\boldsymbol{\theta}} \mathbb{E}_{(\mathcal{K},\boldsymbol{x},y) \sim p(\mathcal{D})} [\mathcal{L}_{\text{CE}}(f_{\hat{\boldsymbol{\theta}}_N^{\mathcal{K}}}(\boldsymbol{x}), y)] \tag{2}$$

where for all $\mathcal{K}$, $n \in \mathbb{N}$, and $\boldsymbol{\theta}$: $\hat{\boldsymbol{\theta}}_0^{\mathcal{K}} = \boldsymbol{\theta}$, and

$$\hat{\boldsymbol{\theta}}_{n+1}^{\mathcal{K}} = \hat{\boldsymbol{\theta}}_n^{\mathcal{K}} - \boldsymbol{\alpha}\nabla\mathcal{L}_{\text{CLM}}(f_{\hat{\boldsymbol{\theta}}_n^{\mathcal{K}}}, \mathcal{K}). \tag{3}$$

Here, $\mathcal{L}_{\text{CE}}(f(\boldsymbol{x}), y)$ denotes the cross-entropy (CE) loss, which we apply with the relevant parameters for each reasoning question in $\mathcal{D}$, $\mathcal{L}_{\text{CLM}}(f, \mathcal{K}) = \frac{1}{|\mathcal{K}|}\sum_{k \in \mathcal{K}}\mathcal{L}_{\text{CLM}}(f, k)$ denotes the causal language modeling loss, and $N$ and $\alpha$ are pre-defined hyperparameters of the fine-tuning. We seek our actual meta-parameters $\boldsymbol{\theta}$ through gradient descent. In particular, denoting by $\boldsymbol{\theta}_0$ our initial meta-parameters, and $\hat{\boldsymbol{\theta}}_{N,i}^{\mathcal{K}}$ the parameters $\hat{\boldsymbol{\theta}}_N^{\mathcal{K}}$ obtained when initializing $\hat{\boldsymbol{\theta}}_0^{\mathcal{K}}$ with $\boldsymbol{\theta}_i$, we iteratively compute

$$\boldsymbol{\theta}_{i+1} = \boldsymbol{\theta}_i - \eta\nabla\frac{1}{|\mathcal{D}_i|}\sum_{(\mathcal{K},\boldsymbol{x},y) \in \mathcal{D}_i}\mathcal{L}_{\text{Total}}(f_{\hat{\boldsymbol{\theta}}_{N,i}^{\mathcal{K}}}, \mathcal{K}, \boldsymbol{x}, y), \tag{4}$$

where $\mathcal{L}_{\text{Total}}(f, \mathcal{K}, \boldsymbol{x}, y) = \mathcal{L}_{\text{CE}}(f(\boldsymbol{x}), y)$ and for each $i$, $\mathcal{D}_i$ is randomly sampled from $p(\mathcal{D})$. This continues until $\mathcal{L}_{\text{Total}}$ converges.

The training can be seen as two nested loops: at each iteration, the **outer loop** (Equation (4)) samples a random batch $\mathcal{D}_i \subseteq \mathcal{D}$ of reasoning problems for evaluating (in order to update) the current meta-parameters $\boldsymbol{\theta}_i$, after the **inner loop** (Equation (3)) adapts them to encode the associated knowledge through $N$ steps of gradient updates.

**Multi-Task Objective**    Through our experiments, we find that adding a knowledge-recovery objective to the outer loop—such that the model must also state all of $\mathcal{K}$ when

---

**Algorithm 1** RECKONING

**Require:** An example distribution $p(\mathcal{D})$, a transformer language model $f$, initial meta-parameters $\boldsymbol{\theta}$, outer step size $\eta$, inner step size $\boldsymbol{\alpha}$, inner loop length $N$.
1: **while** not converged **do**           ▷ outer loop
2:     Sample $\mathcal{D}' \sim p(\mathcal{D})$
3:     $\mathcal{L}_{\mathcal{D}'} \leftarrow 0$
4:     **for** each $(\mathcal{K}, \boldsymbol{x}, y) \in \mathcal{D}'$ **do**
5:         Initialize $\hat{\boldsymbol{\theta}}_0^{\mathcal{K}} = \boldsymbol{\theta}$
6:         **for** $n := 0$ **to** $N - 1$ **do**        ▷ inner loop
7:             $\hat{\boldsymbol{\theta}}_{n+1}^{\mathcal{K}} \leftarrow \hat{\boldsymbol{\theta}}_n^{\mathcal{K}} - \boldsymbol{\alpha}\nabla\mathcal{L}_{CLM}(f_{\hat{\boldsymbol{\theta}}_n^{\mathcal{K}}}, \mathcal{K})$
8:         **end for**
9:         $\mathcal{L}_{\mathcal{D}'} \leftarrow \mathcal{L}_{\mathcal{D}'} + \mathcal{L}_{\text{Total}}(f_{\hat{\boldsymbol{\theta}}_N^{\mathcal{K}}}, \mathcal{K}, \boldsymbol{x}, y)$
10:     **end for**
11:     $\boldsymbol{\theta} \leftarrow \boldsymbol{\theta} - \eta\nabla\frac{1}{|\mathcal{D}'|}\mathcal{L}_{\mathcal{D}'}$
12: **end while**

---

prompted with $\boldsymbol{x}$—improves the model's reasoning performance. We evaluate knowledge recovery with a CLM loss and combine the two losses by simple addition, following prior works [24, 77, 78]. The entire change is achieved by redefining the total loss in our outer loop (Equation (4)) as:

$$\mathcal{L}_{\text{Total}}(f, \mathcal{K}, \boldsymbol{x}, y) = \mathcal{L}_{\text{CE}}(f(\boldsymbol{x}), y) + \mathcal{L}_{\text{CLM}}(f, \boldsymbol{x}, \mathcal{K}) \tag{5}$$

where $\mathcal{L}_{\text{CLM}}(f, \boldsymbol{x}, \mathcal{K})$ is the language modeling loss on $\mathcal{K}$, as in Equation (3), but this time conditioned on the question $\boldsymbol{x}$. The overall process for training RECKONING is depicted in Algorithm 1 and Figure 2. Additionally, we dynamically learn a per-step-per-layer learning rate to replace the shared constant learning rate in the inner loop. We give more details in Appendix D.

## 4   Experiments

**Setup**    We conduct our experiments on three datasets focusing on multi-hop logical reasoning over natural language knowledge: **ProofWriter** [74], which measures the model's ability to emulate reasoning over facts and rules expressed in natural language; **CLUTRR-SG** [29], which is generated from the CLUTRR [72] benchmark, a logical reasoning task that involves reasoning over family relationships between entities grounded in first-order logical proofs; and **FOLIO** [30], a reasoning benchmark with first-order logical reasoning problems written by expert annotators based on real-world knowledge. Each problem in these datasets requires multiple reasoning hops to answer.[1]

---

[1]In ProofWriter, the number of reasoning hops is called the proof depth. To unify the presentation of the results, we use the term "hop" to describe the number of reasoning steps for both datasets.

We compare our method against the following baselines: (1) a fine-tuned model that performs a forward pass on only the question without access to the knowledge (**No-Facts**), (2) a fine-tuned model that performs a forward pass on only the knowledge without access to the question (**No-Question**), (3) a model trained using RECKONING with random knowledge that is not relevant to the questions (**Random-Facts**), and (4) an ICR baseline that concatenates the knowledge $\mathcal{K}$ with the question $x$ in a single context and is trained using supervised learning to predict the answer (**FT-ICR**). Our first three baselines sanity-check whether any surface-level patterns in the questions and facts can be exploited to make accurate predictions. The last baseline compares RECKONING to the conventional way of reasoning with language models. Unless stated otherwise, we use the GPT-2-small [60] model ($\sim$124M parameters) as our initialization and refer by RECKONING to our method trained with the multi-task objective. We compute each score from the average across three different runs. For more details on the implementation, datasets, and examples, see Appendix A and Appendix C.

### 4.1 Multi-hop Reasoning Performance

**Main Results** We first evaluate whether RECKONING learns to perform reasoning in the base setting. A model is given a set of supporting facts (without distractors) and a question (or hypothesis) as input and begins by performing a few CLM learning steps on the facts. Then, the updated model reads **only** the question and generates an answer. To answer correctly, the model must reason over both facts and the question, meaning it must encode the facts during the inner loop such that multi-hop reasoning can be performed over them later.

| | ProofWriter | | | CLUTRR-SG | | |
|---|---|---|---|---|---|---|
| Method | 2-h | 3-h | 5-h | 2-h | 4-h | 6-h |
| No-Facts | 64.1 | 63.0 | 64.2 | 0.0 | 8.8 | 8.9 |
| No-Question | 66.2 | 67.0 | 65.2 | 35.7 | 36.4 | 28.7 |
| Random-Facts | 64.1 | 63.0 | 64.2 | 0.0 | 1.3 | 2.5 |
| FT-ICR$_{ST}$ | 98.4 | 98.8 | 97.8 | 97.4 | 91.3 | 89.1 |
| FT-ICR$_{MT}$ | 99.4 | 99.2 | 99.6 | 98.1 | 96.9 | 90.3 |
| RECKONING$_{ST}$ | 98.3 | 98.3 | 99.1 | 96.0 | 90.2 | 91.2 |
| RECKONING$_{MT}$ | **99.5** | **99.7** | **99.8** | **98.3** | **97.6** | **94.8** |

Table 1: Label accuracy of RECKONING on ProofWriter and CLUTRR-SG, compared to FT-ICR baselines where the supporting facts are given as part of the input. MT marks models trained with the multi-task objective, which optimizes both question-answering and knowledge memorization.

We train our models and the fine-tuned ICR (FT-ICR) baselines with both the single-task ($\mathcal{L}_{CE}$) and multi-task ($\mathcal{L}_{CE} + \mathcal{L}_{CLM}$) objectives. For multi-task (MT) training, the model learns to answer the question and generate its relevant knowledge in the outer loop. Table 1 shows the evaluation results on question answering (or hypothesis classification). For all hop numbers in ProofWriter and CLUTRR-SG, multi-task RECKONING outperforms the best result of all baselines (consistently obtained by multi-task FT-ICR) by an average of $1\%$. We conclude that RECKONING can effectively solve reasoning problems through its updated parametric knowledge and do so better than existing baselines. The multi-task objective is crucial for this success: not only is RECKONING's performance consistently higher (by an average of $2.8\%$ over the two datasets and their hop counts) when using the multi-task rather than single-task (ST) objective, but it also under-performs both FT-ICR baselines when trained with only the single-task objective. The multi-task objective also improves FT-ICR consistently (average $1.8\%$), though it is not enough to beat the multi-task RECKONING. In all further experiments, we consider only RECKONING and FT-ICR with a multi-task objective.

**Generalizing to Longer Reasoning Chains** Our first experiments assume an alignment between the number of reasoning hops in the questions in the training and test set. However, we may not be able to train on all *n*-hop reasoning questions we encounter in the wild, and we rarely know the number of reasoning hops in a question *a priori*. Consequently, we also measure the generalization capacity of our model to questions with hop numbers unseen during training. We compile *interpolation* (fewer hops than the train set) and *extrapolation* (more hops than the train set) test sets from the CLUTRR-SG dataset. Again, we train models individually on 2-hop, 4-hop, and 6-hop examples and evaluate these three sets of models on the test sets, which contain 2-10-hop reasoning questions. Figure 3 shows that both RECKONING models and ICR baselines retain high performance on the interpolation test sets but exhibit decreasing performance as the number of hops increases. Importantly, though, RECKONING outperforms FT-ICR on all test sets regardless of the number of training hops, with the highest difference being more than 10% in every training setting (15%, 30%, 10%, respectively). These performance gains when testing on extrapolation data suggest that training with RECKONING better generalizes to examples with high OOD hop counts than in-context reasoning (ICR).

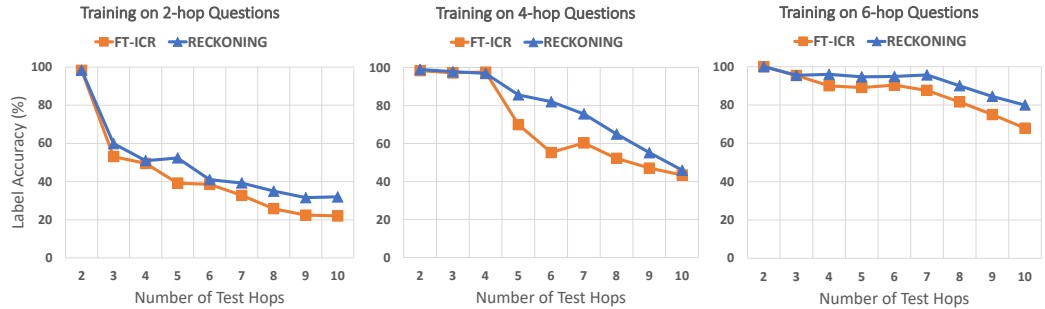

Figure 3: System generalization evaluation on CLUTRR-SG. From left to right, the models are trained on 2-hop, 4-hop, and 6-hop CLUTRR-SG data portions. We evaluate the model on 2-10 hop test sets. The higher the hops, the more facts a question has, and the more difficult that question is.

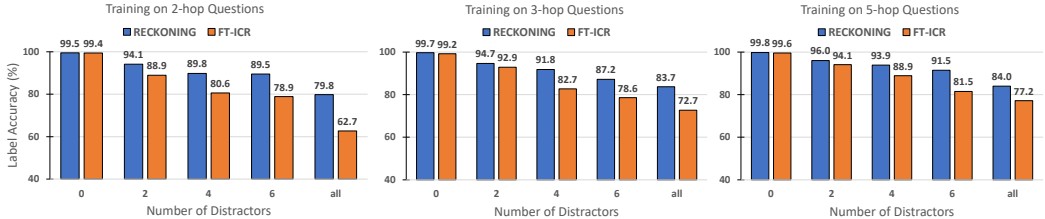

Figure 5: Robustness under distractors for ProofWriter. Each of the three plots corresponds to training and testing on a subset of questions in ProofWriter with a different number of hops (2,3,5-hops). Each bar corresponds to the number of distractors in the knowledge sets for those questions.

**Does RECKONING's performance depend on the number of inner loop gradient steps?** In RECKONING, the model performs multi-hop reasoning over facts by encoding facts using multiple gradient steps in the inner loop optimization (§3). Naturally, this process prompts the question of whether there is a correlation between the number of reasoning hops and the number of gradient steps needed to reliably encode the knowledge (i.e., problems with more reasoning hops require more gradient steps in the inner loop to encode the facts). In Figure 4, we show for CLUTRR-SG that as the number of inner loop steps increases, the label accuracy of the outer-loop task also increases. Furthermore, when considering the performance gains for reasoning with 6 inner loop steps (i.e., knowledge encoding) as opposed to one, we observe that this gap is much more pronounced for 4-hop (42.3%) and 6-hop (34.7%) reasoning than it is for 2-hop reasoning (5.9%). These results show that problems requiring more hops of reasoning

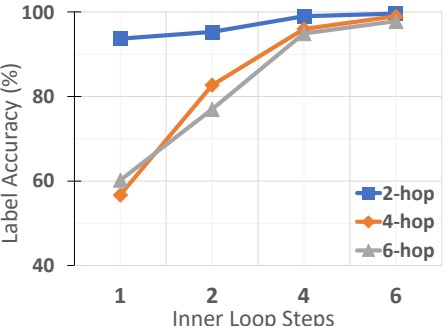

Figure 4: Multi-hop reasoning performance as a function of the number of inner loop steps (x-axis), with each line focusing (by training and testing) on CLUTRR-SG with a different number of hops.

also greatly benefit from more steps of inner loop knowledge encoding.

### 4.2 Reasoning with Distractors

In cases where multiple questions must be answered about the same knowledge set, some knowledge that is relevant to one question will likely be irrelevant to another question. For example, in Table 7, the fact "Charlie is White." is not needed to answer the question "Harry is red?". Thus, it is important to evaluate the robustness of RECKONING when there exists irrelevant information (i.e., distractors) in the knowledge set. In this experiment, we analyze RECKONING's ability to focus on the correct knowledge and ignore distractors when answering questions. We use ProofWriter as the evaluation dataset since it already has a setting with distractors included in the knowledge. For systematic analysis, we gradually add distractors to the context (starting from 2 and finishing at *all* possible

distractors, of which there are an average of 7 per question). We train RECKONING and the baseline using the multi-task objective, where the model must (1) recall all of the facts and rules relevant to the question and (2) predict the conclusion based on the correct knowledge. In this case, we adapt training such that for each question $x$, the outer-loop (Equation (5)) CLM loss is only computed with respect to the relevant facts from $\mathcal{K}$, thereby learning to recall only relevant facts during training.

In Figure 5, we see that RECKONING's performance is consistently more robust under distractors than the FT-ICR baseline. When we include all of the distractors in the context, RECKONING achieves a significantly higher average label accuracy (82.5%) across hops than the baseline (70.9%), as computed by the average of the 3 considered hop depths. Additionally, compared to performance with no distractors, RECKONING's performance only drops 17.1% while the baseline performance drops 28.6%, thereby exhibiting a better ability to disentangle the correct knowledge from the distractors.

Finally, we also explore RECKONING's generalizability to models with a larger parameter size. We scale up the language model we used, GPT-2-small (124M), to GPT-2-XL (1.5B) by adopting a parameter efficient finetuning method *LoRA* [34]. For simplicity, we only evaluate the models on the most difficult settings, i.e., ProofWriter-5-hop with all the distractors. With GPT-2-XL-LoRA, in-context reasoning achieves 65% accuracy on the test set, while our RECKONING model achieves 70.2% accuracy, a 5% performance gain. This result suggests that RECKONING's advantages in the presence of distractors hold even as models scale in size.

### 4.3 Generalization to Real-World knowledge

To investigate how generalizable our method is to real-world knowledge beyond the synthetic setting, we evaluate RECKONING on a more real-world multi-hop logical reasoning task, FOLIO [30], and report the result in Table 2. The dataset has a rich vocabulary, diverse logic patterns, and abundant language variations. It has been shown to challenge LLMs in both supervised fine-tuning and in-context learning settings. We fine-tune the GPT-2 model following the in-context reasoning setting as the baseline. As before, we train the GPT-2 model and RECKONING using the multi-task objective. We also compare to more advanced baselines, including GPT-3.5 (text-davinci-003 [56]) and ChatGPT(gpt-3.5-turbo [2]), two popular large language models with around 175B parameters. For these two large models, we evaluate both in the zero-shot and few-shot settings. In the few-shot setting, we prompt the model with 8 single-task examples randomly sampled from the training set to perform in-context learning. We find that RECKONING's performance (which is initiated here from GPT-2) is better than the GPT-2 in-context reasoning baseline. Compared to the two advanced large language models, RECKONING outperforms them by a significant margin (12% 0-shot and 7% 8-shot). We conclude that RECKONING is effective and significantly benefits reasoning tasks using real-world knowledge.

| Model | Acc. |
|---|---|
| Random | 33.3 |
| GPT-2 | 53.3 |
| GPT-3.5 (text-davinci-003) 0-shot | 45.1 |
| GPT-3.5 (text-davinci-003) 8-shot | 52.9 |
| ChatGPT (gpt-3.5-turbo) 0-shot | 40.0 |
| ChatGPT (gpt-3.5-turbo) 8-shot | 42.6 |
| RECKONING | **54.9** |

Table 2: Evaluation results on FOLIO. We compare RECKONING against the FT-ICR baseline with GPT-2 and two popular large language models.

### 4.4 Run-time Analysis

One of the advantages of RECKONING is the ability to memorize a large set of knowledge $\mathcal{K}$ and answer multiple related questions about that knowledge at a little extra cost per question. Specifically, in contrast to ICR, RECKONING can encode $\mathcal{K}$ once and answer multiple questions without needing to reprocess it for each question asked. To test whether RECKONING could be a more efficient method for inference in this setting, we measure the wall-clock time (in seconds) of the complete inference pipeline of RECKONING vs. ICR. For this experiment, we use a synthetic reasoning dataset in which $\mathcal{K}$ is a sequence of random letters, and the question $x$ asks for the most frequent letter in the context. The total number of tokens in each example is 1024: 7 for $x$, 1 for the answer, and the remaining 1016 for $\mathcal{K}$, broken into 8 "facts".

---

[2]https://openai.com/blog/chatgpt

The FT-ICR baseline receives a sequence including all 8 facts and the question. In contrast, RECKONING receives the 8 facts as a batch of eight segments of 127 tokens and encodes them in parallel in the inner loop. In the outer loop, the model only receives the question or a batch of questions. We focus on two settings: (1) inference time for a single question and (2) inference time when answering multiple questions. In the multiple-question setting, we set the number of questions to 18 (the same as in ProofWriter). For RECKONING, the inference process includes the inner-loop knowledge encoding and the final forward pass to encode the question. We set the number of inner loop gradient steps to 1 and 4. In Table 3, we see that when answering a single question, RECKONING does not perform inference faster than in-context reasoning. However, RECKONING shows significant advantages under a multi-question setting. Both the 1-step inner loop and the 4-step inner loop are faster than the baseline. Since RECKONING encodes the knowledge in model parameters, it does not need to reprocess the knowledge for a related question and is more efficient. We run this experiment on 1 RTX 3090 GPU.[3]

| Model | Wall-clock Time (s) |
|---|---|
| *Single question* | |
| FT-ICR | 0.1887 |
| RECKONING$_{1step}$ | 0.2532 |
| RECKONING$_{4step}$ | 0.9664 |
| *Multiple questions (18)* | |
| FT-ICR | 2.0436 |
| RECKONING$_{1step}$ | 0.6228 |
| RECKONING$_{4step}$ | 1.4839 |

Table 3: Wall clock run-time, in seconds, of the fine-tuned ICR baseline and RECKONING.

## 4.5 Memorizing Knowledge

In Table 1, we saw that training RECKONING with a multi-task (MT) outer loop objective improved over training with the single-task (ST) objective, potentially because the MT objective improves the model's ability to memorize the knowledge in the inner loop. To validate our hypothesis, we analyze RECKONING's performance in reproducing memorized knowledge. First, we show in Table 4 the inner loop average loss

| Method | $\mathcal{L}_{CLM}$ | $\Delta\mathcal{L}_{CLM}$ |
|---|---|---|
| RECKONING$_{ST}$ | 10.74 | 9.55 |
| RECKONING$_{MT}$ | 0.167 | 12.61 |

Table 4: Average inner loop validation loss: final ($\mathcal{L}_{CLM}$) and difference from start to finish ($\Delta\mathcal{L}_{CLM}$).

($\mathcal{L}_{CLM}$) and average change ($\Delta\mathcal{L}_{CLM}$) (from first inner loop evaluation to last) on validation examples from the 5-hop ProofWriter data. We see that the average inner loop loss for RECKONING$_{ST}$ is much higher than RECKONING$_{MT}$, and indeed starts out much higher as well. This shows that the ST outer loop objective, which optimizes the model only for question answering, does not learn to encode the knowledge in the inner loop by *memorizing* it. In contrast, the MT objective forces the model to learn to memorize the knowledge, too: we observe that RECKONING$_{MT}$ minimizes the inner loop loss as it processes the knowledge. This pattern is also shown in the average inner-loss difference ($\Delta\mathcal{L}_{CLM}$): the inner loop loss decreases more after the gradient updates when trained with the MT objective. Next, we report in Table 5 the model's ability to reproduce memorized facts correctly under a multi-task setting, as measured by an exact match score between the reproduced facts and the gold facts. [4] We evaluate on the ProofWriter dataset both with and without distractors in the context and compare the results to the FT-ICR baseline.

| | ProofWriter | | | ProofWriter$_{distractor}$ | | |
|---|---|---|---|---|---|---|
| Method | 2-h | 3-h | 5-h | 2-h | 3-h | 5-h |
| FT-ICR$_{MT}$ | **99.8** | **99.0** | **98.7** | 42.3 | 50.3 | 55.6 |
| RECKONING$_{MT}$ | 98.9 | 98.6 | 98.2 | **71.2** | **74.4** | **75.1** |

Table 5: Exact match score for reproducing memorized knowledge. In contrast to in-context reasoning, RECKONING does not have direct access to the knowledge.

The results show that RECKONING$_{MT}$ can successfully (average exact match score of 99.3%) recover the relevant facts from its model parameters when the context does not include any distractors. Note that this is comparable to the FT-ICR baseline, for which the task is much easier as it can directly attend to and copy the facts from input, while

RECKONING$_{MT}$ no longer has direct access to them. When the context includes distractors, both RECKONING and FT-ICR struggle to identify and reproduce *only* the relevant facts. However, the per-

---

[3]We perform this experiment in a limited setting and do not handle the case where hidden states could be cached for the forward pass of in-context reasoning, likely speeding up multi-question inference [12].

[4]This is done by prompting the model with the question and comparing its output (after its answer to the question) to the concatenation of all facts. The model is able to produce these facts in the expected order due to an implementation detail: they are numbered and labeled when given to the inner loop.

formance for FT-ICR (average $49.4\%$) drops far below that of RECKONING ($73.6\%$), demonstrating that RECKONING is much better at disentangling the relevant knowledge from the distractors.

Finally, we show that RECKONING with a multi-task objective is also more robust to distractors as it trains the model to reproduce only the facts that would be relevant to a particular question we ask in the outer loop. As in Section 4.2, we use the ProofWriter dataset and, for each question, add all the distractors to the context. We train the model using the multi-task objective and report the label accuracy. While in Table 1, we originally saw a $\sim 1\%$ improvement from training with a multi-task objective on ProofWriter with no distractors, we see a much more significant performance gap in Figure 6 ($\sim 18.2\%$) when distractors are available. We also note that the performance of the single-task model is essentially *random* (see the Random-Facts baseline from Table 1). By learning *how* to memorize knowledge in the inner loop to recall relevant facts in the outer loop, the model also learns how to encode facts more robustly over them.

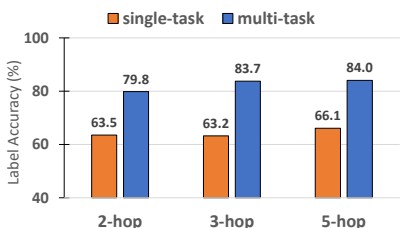

Figure 6: RECKONING performance when trained with a single-task and a multi-task objective under distractors. When trained with the multi-task objective, the model learns to memorize and reason over the relevant facts.

## 5   Related Work

**Logical Reasoning Datasets and Benchmarks**   As a central building block of human cognition and intelligence [28], logical reasoning has been a long-pursued topic in the field of AI [2, 9, 13, 17, 46, 48, 55, 73]. Logical reasoning, in general, can be categorized in a trichotomy of deductive, inductive, and abductive reasoning [26]. Multiple datasets have been published that evaluate neural models' ability on these three types of logical reasoning [6, 17, 72]. Initially, logical reasoning tasks focused on hypothesis classification, where, given a theory consisting of multiple facts and rules, a model would determine whether the hypothesis was correct. Recently, transformer-based language models have been directly used to solve this task in synthetic [17, 66], real-world [30], and adversarial [27, 62, 68] settings. However, simply predicting whether the hypothesis is valid does not elucidate whether the model correctly reasons over the provided knowledge. To better analyze and interpret the reasoning process of language models, new tasks focus on generating the valid proof that explains the model's decision [20, 74]. Our proposed method, RECKONING, is optimized for the hypothesis classification reasoning task and evaluates on many of these datasets [29, 74, 30].

**Logical Reasoning over Natural Language**   Historically, automatic logical reasoners used symbolic systems and formal languages as a knowledge representation [1, 43, 49, 52, 55, 81]. However, these systems were hard to scale up due to the knowledge-acquisition bottleneck and the brittleness of formal representation [35, 84]. With recent advances in transformer-based language modeling [76] and self-supervised pre-training [21, 60, 61], a novel paradigm for logical reasoning emerged, where pre-trained language models (PLMs) could be used as soft reasoners over knowledge expressed in natural language. Natural language as a knowledge representation allowed PLMs to handle raw input with diverse formats [15, 32], resulting in PLMs being applied to various types of deductive [17], abductive [6], and inductive [29] reasoning tasks. However, language models as soft reasoners also showed structural weaknesses, as their performance dropped on complex logical operations [13, 80], and their reasoning process was not interpretable [45, 65]. Consequently, a new line of work uses neuro-symbolic methods to combine the best of both language models and symbolic reasoning [7, 14, 36, 41, 44]. Specifically, the interpretability gap motivated modular and step-wise reasoning systems that use PLMs as intermediate modules [33, 59, 67, 69, 75, 83] to generate reasoning steps (e.g., proofs). In contrast to these works, our method RECKONING dynamically encodes natural language knowledge into the model parameters, thereby reasoning by mixing contextual knowledge with pre-encoded parametric knowledge and allowing the model to determine a conclusion based on its updated parametric knowledge.

**Model Editing**   While our motivations are grounded in research on machine reasoning, our methods are more often used in the area of model editing. Model editing is a method to edit a model's parameters to correct its errors or update the model. Several works propose hypernetwork-based methods to edit knowledge in a model by predicting updates conditioned on new factual statements [31] or transforming the gradients from new provided facts [54] to make local edits to a model. Other

approaches focus on more direct edits of model behavior, such as directly modifying neuron outputs [19, 85], localizing distinct feed-forward layers that are responsible for factual recall, and modifying these weights [50], and performing weight updates across multiple layers to perform simultaneous edits [51]. Similarly, our method also rapidly edits the model parameters to add knowledge. However, our bi-level framework optimizes model edits for the reasoning task in the outer loop, allowing the model to learn to quickly memorize knowledge that can support the model's reasoning ability.

**Language Models as Knowledge Bases**   Our work learns to reason by dynamically encoding contextual knowledge in the parameters of language models before answering questions about them. Previous studies have found that LLMs can store real-world facts learned during pre-training [5, 10, 11, 50, 64]. Learning these facts during pre-training allows language models to be prompted [39, 58, 71, 86] or adapted [8, 37, 38, 63] to produce these facts on-demand. However, LLM knowledge is latent and hard to identify or control. The model generation is sensitive to specific words or phrases. LLMs emit knowledge encoded in the parameters only when prompted appropriately [10, 18, 23, 57]. It is also difficult to inject or update knowledge for LLMs [50], and the memorization of knowledge in LLMs is not optimized toward their reasoning ability. In our work, we seek to find a way to add knowledge to LLMs in a controllable and adaptive way that can benefit downstream reasoning applications.

## 6   Conclusion

We present RECKONING, a bi-level learning framework for multi-hop reasoning that encodes knowledge verbalized using natural language into a model's parameters through gradient updates. During training, the inner loop encodes the contextual knowledge into the model parameters by backpropagating a language modeling loss. In the outer loop, given only the question as input, the model solves reasoning problems using the memorized knowledge. Through bi-level optimization, RECKONING finds a set of meta-parameters that allows it to perform quick knowledge-based updates for reasoning. Our experiments show that RECKONING learns to reason only by relying on its parametric knowledge after the external knowledge has been encoded. Using a multi-task objective that jointly optimizes reasoning and knowledge memorization in the outer loop, RECKONING outperforms ICR baselines that are trained to encode external knowledge as part of the context. Through our analysis, we show that RECKONING is more generalizable to problems with longer reasoning chains, less susceptible to irrelevant distractor knowledge, and that RECKONING is more efficient than the baseline when answering multiple questions that require common knowledge.

## Acknowledgements

We thank Shikhar Murty and Christopher Manning for helpful discussions in crafting ideas for this project. We also gratefully acknowledge the support of the Swiss National Science Foundation (No. 215390), Innosuisse (PFFS-21-29), the EPFL Science Seed Fund, the EPFL Center for Imaging, Sony Group Corporation, and the Allen Institute for AI.

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

## A  Dataset

**ProofWriter**  The ProofWriter [74] dataset has 500k pairs of questions, answers, and proofs over natural-language rule bases. Each example in the dataset contains a set of facts, a set of rules, a hypothesis, and a label indicating whether the hypothesis is true, false, or unknown. The dataset comprise five datasets named D0, D1, D2, D3, D5, each with 100k examples. Each dataset's questions require reasoning up to depths $D$ ($D = 0, 1, 2, 3, 5$) to determine their answers. In our experiments, we only focus on the datasets that require more reasoning depths (D2, D3, D5). We show an example from the dataset in Table 7. In these datasets, a set of facts and rules are mapped to 18 questions, where the questions can be answered based on a subset of the facts and rules. Thus, some of the facts or rules can be irrelevant to some questions, and we call them distractors in Section 4.2. In the experiment for knowledge encoding with distractors, we encode all the facts in the model parameters and evaluate its ability to reproduce and reason over the correct facts. We show an example of distractor and relevant knowledge of a question in Table 9. For detailed statistics on the two datasets, please see Table 6.

**CLUTRR-SG**  The CLUTRR-SG [29] is an evaluation dataset for inductive reasoning on family relations adapted from the [72] dataset for measuring systematic generalization. Each example in the dataset contains (i) a set of facts representing a family graph $G = (V, E)$ where nodes ($V$) are entities and edges ($E$) are the relationships. (ii) a question asking the relationship between two entities ($v_1, v_n \in V$), and (iii) a target relationship $e^* \in E$ as the answer for the question. The facts are expressed as a list of ($v_i, e_j, v_k$) tuples. The two entities in the question are separated by more than one hop in the graph. There are 272 unique entities, 20 relationship types, and nearly 1.5M possible facts in the dataset. Following the authors, we define the difficulty of examples based on the number of family graph edges (i.e., the number of reasoning hops required to determine a relation), in which $k$ edges ($k$-hop) correspond to $k$ facts. We show an example from the dataset in Table 8.

## B  In-context Reasoning with Distractors

To motivate the advantage of RECKONING on mitigating interference from distractors, we analyze the performance change of fine-tuned in-context reasoning with and without distractors present in the context of the questions. We define distractors as additional facts or rules present in a question's context that are not directly relevant to the questions. A model should not be able to use only these distractors to answer a question correctly. For an example of distractors in a question's context, please see Table 9. We evaluate the baseline on the ProofWriter dataset since it naturally contains contexts including distractors (Table 9). Recall that we have two training objectives. The single-task objective only trains the model to predict an answer for each question given their contexts. The

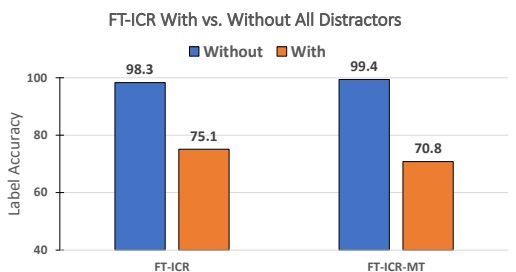

Figure 7: Label accuracy of fine-tuned in-context reasoning on questions with and without distractors in the context. With the same questions, adding distractors to contexts significantly lowers the performance of in-context reasoning, both in the single-task and multi-task settings.

multi-task objective (MT) trains the model not only to predict an answer but also to reproduce the correct facts and rules (in contrast to distractors) based on the contexts. We evaluate the baseline on 2, 3, and 5-hop datasets with both training objectives, and we report the average label accuracy across hops in Figure 7. Compared to the baseline's performance without distractors in the context, the performance with distractors decreases significantly. For single-task, the performance drops 23.2% when adding distractors to the contexts, and the performance with the multi-task objective drops 28.6%. The results highlight in-context reasoning's high sensitivity to the interference of irrelevant information in the contexts.

## C  Implementation Details

We select GPT-2-base [60] as the model for our method and all the baselines. We use the version implemented by the Huggingface Transformers library [79]. All the experiments for RECKONING

| Dataset | #Train | #Validation | #Test |
|---|---|---|---|
| CLUTRR-SG (2-hop) | 96,012 | 10,972 | 3,102 |
| CLUTRR-SG (4-hop) | 89,972 | 10,086 | 9,946 |
| CLUTRR-SG (6-hop) | 90,922 | 10,290 | 8,788 |
| ProofWriter (2-hop) | 6,996 | 1,098 | 2,013 |
| ProofWriter (3-hop) | 10,854 | 1,641 | 3,057 |
| ProofWriter (5-hop) | 18,525 | 2,553 | 5,175 |

Table 6: Dataset splits and statistics for our experiments

| Identifier | Content |
|---|---|
| fact 1 | Harry is nice. |
| fact 2 | Fiona is quite Nice. |
| fact 3 | Fiona is round. |
| fact 4 | Fiona is white. |
| fact 5 | Dave is furry. |
| fact 6 | Charlie is white. |
| rule 1 | Furry people are green. |
| rule 2 | Round, green people are red. |
| rule 3 | All red people are white. |
| rule 4 | Nice, round people are furry. |
| rule 5 | If someone is nice, then they are round. |
| rule 6 | If Charlie is round and Charlie is nice, then Charlie is white. |
| question-answer 1 | Harry is red? True |
| question-answer 2 | Harry is not red? False |
| question-answer 3 | Dave is not white? Unknown |

Table 7: An example from the dataset ProofWriter. There are 6 facts and 6 rules mapped to three question-answer pairs. Each question can be answered based on the given facts and rules.

are conducted on a cluster with NVIDIA A100 (40GB) GPUs. All the baseline experiments are conducted on a local machine with NVIDIA RTX 3090 GPU (24GB).

**Fine-tuned In-context Reasoning** We set the train batch size to 16 and train the model for 6 epochs with early stopping based on the validation label accuracy. We set the learning rate to 3e-5 and use the AdamW optimizer with $\epsilon$ set to 1e-8. We validate the model on the development set for every epoch and select the best checkpoint using the validation accuracy as the metric.

RECKONING In the inner loop, we generally perform 4 gradient steps for lower-hop questions (2, 3, 4-hop) and 5 gradient steps for higher-hop questions (5 and 6-hop). We select the AdamW [47] as the optimizer for the inner loop since the main task is language modeling. The inner-loop learning rate is set to 3e-5 before training, and the algorithm dynamically learns a set of optimal learning rates when converged. In our experiments and analysis, we only report the results from RECKONING with a multi-task objective since its performance is better than the single-task objective. In the outer loop, we also use the AdamW with a learning rate of 3e-5. For both optimizers, we set $\epsilon$ to 1e-8. We set the train batch size to 2 due to memory limitations. We apply the technique of gradient accumulation and set the accumulation step to 2. We train the model for 6 epochs with early stopping. For each epoch, we validate the model twice: once in the middle and once at the end. We select the best model checkpoint based on the validation label accuracy.

## D   Adaptive Learning Rate

Prior works [3, 4] show that a fixed learning rate shared across steps and parameters does not benefit the generalization performance of the system. Instead, [3] recommends learning a learning rate for

| Identifier | Content |
| --- | --- |
| fact 1 | C is H's father. |
| fact 2 | Z is J's aunt. |
| fact 3 | J is S's daughter. |
| fact 4 | D is C's father |
| fact 5 | S is B's father. |
| fact 6 | H is Z's son. |
| question-answer 1 | How are D and B related to each other? Grandfather |

Table 8: An example of 6-hop reasoning from the CLUTRR-SG dataset.

| Identifier | Content |
| --- | --- |
| fact 1 | Harry is nice. |
| fact 2 | Fiona is quite Nice. |
| fact 3 | Fiona is round. |
| fact 4 | Fiona is white. |
| fact 5 | Dave is furry. |
| fact 6 | Charlie is white. |
| rule 1 | Furry people are green. |
| rule 2 | Round, green people are red. |
| rule 3 | All red people are white. |
| rule 4 | Nice, round people are furry. |
| rule 5 | If someone is nice, then they are round. |
| rule 6 | If Charlie is round and Charlie is nice, then Charlie is white. |
| question-answer 1 | Harry is red? True |

Table 9: Example of distractors (black) and relevant knowledge (red) in the ProofWriter dataset.

each network layer and each adaptation step in the inner loop. The layer parameters can learn to adjust the learning rates dynamically at each step. To control the learning rate $\boldsymbol{\alpha}$ in the inner loop adaptively, we define $\boldsymbol{\alpha}$ as a set of adjustable variable: $\boldsymbol{\alpha} = \{\boldsymbol{\alpha}_0, \boldsymbol{\alpha}_1, ...\boldsymbol{\alpha}_L\}$, where $L$ is the number of layers and for every $l = 0, ..., L$, $\boldsymbol{\alpha}_l$ is a vector with $N$ elements given a pre-defined inner loop step number $N$. The inner loop update equation then becomes

$$\hat{\boldsymbol{\theta}}_{n+1,l}^{\mathcal{K}} = \hat{\boldsymbol{\theta}}_{n,l}^{\mathcal{K}} - \boldsymbol{\alpha}_n^{(l)} \odot \nabla \mathcal{L}_{\text{CLM}}(f_{\hat{\boldsymbol{\theta}}_n^{\mathcal{K}}}, \mathcal{K}) \tag{6}$$

where $\odot$ is an element-wise product and $\hat{\boldsymbol{\theta}}_n^{(l)}$ is the parameters for layer $l$ at the inner step $n$. We learn the set of optimal inner loop learning rates $\boldsymbol{\alpha}^*$ by optimizing the parameters in the outer loop:

$$\boldsymbol{\alpha} \leftarrow \boldsymbol{\alpha} - \eta \nabla \frac{1}{|\mathcal{D}_i|} \sum_{(\mathcal{K}, \boldsymbol{x}, y) \in \mathcal{D}_i} \mathcal{L}_{\text{Total}}(f_{\hat{\boldsymbol{\theta}}_{[}^{\mathcal{K}} \boldsymbol{\theta}_i]N}(\boldsymbol{x}), y), \tag{7}$$

where $\eta$ is the outer loop learning rate and $\hat{\boldsymbol{\theta}}$ is the updated parameters from inner loop. Below, we show the final algorithm of RECKONING in Algorithm 2.

**Algorithm 2** Dynamic Knowledge Encoding for Reasoning

---

**Require:** An example distribution $p(\mathcal{D})$, a transformer language model $f$, initial meta-parameters $\boldsymbol{\theta}$, outer step size $\eta$, initial inner step size $\boldsymbol{\alpha}$, inner loop length $N$.

1: **while** not converged **do**             ▷ outer loop
2:     Sample $\mathcal{D}' \sim p(\mathcal{D})$
3:     $\mathcal{L}_{\mathcal{D}'} \leftarrow 0$
4:     **for** each $(\mathcal{K}, \boldsymbol{x}, y) \in \mathcal{D}'$ **do**
5:        Initialize $\hat{\boldsymbol{\theta}}_0^{\mathcal{K}} = \boldsymbol{\theta}$
6:        **for** $n := 0$ **to** $N-1$ **do**            ▷ inner loop
7:           $\hat{\boldsymbol{\theta}}_{n+1}^{\mathcal{K}} \leftarrow \hat{\boldsymbol{\theta}}_n^{\mathcal{K}} - \boldsymbol{\alpha} \odot \nabla \mathcal{L}_{CLM}(f_{\hat{\boldsymbol{\theta}}_n^{\mathcal{K}}}, \mathcal{K})$
8:        **end for**
9:        $\mathcal{L}_{\mathcal{D}'} \leftarrow \mathcal{L}_{\mathcal{D}'} + \mathcal{L}_{\text{Total}}(f_{\hat{\boldsymbol{\theta}}_N^{\mathcal{K}}}, \mathcal{K}, \boldsymbol{x}, y)$
10:     **end for**
11:     $\boldsymbol{\alpha} \leftarrow \boldsymbol{\alpha} - \eta \nabla \mathcal{L}_{\mathcal{D}'}$           ▷ Update inner step size
12:     $\boldsymbol{\theta} \leftarrow \boldsymbol{\theta} - \eta \nabla \mathcal{L}_{\mathcal{D}'}$
13: **end while**

---

**Are dynamic learning rates necessary for RECK-ONING's performance?** Following prior works on meta-learning [3, 4], we dynamically learn a set of per-step-per-layer learning rates for RECKONING. In this ablation study, we analyze whether dynamic learning rates for the inner loop effectively improve the outer loop reasoning performance. Similarly, we fix other experimental settings and set the number of inner loop steps to 4. As Figure 8 shows, when using a static learning rate (i.e., all layers and inner loop steps share a constant learning rate), the performance drops by a large margin (average drop of 34.2%). The performance drop becomes more significant on questions requiring more reasoning hops (45.5% drop for 4-hop and 39.5% drop for 6-hop), demonstrating the importance of using a dynamic learning rate in the inner loop of our framework.

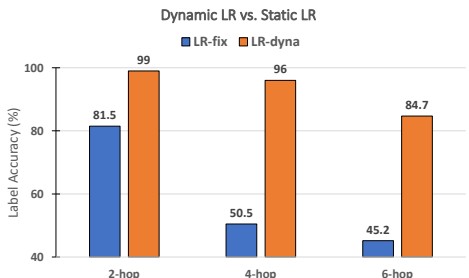

Figure 8: We study how much the dynamic learning rate in the inner loop contributes to the outer loop performance. We fix all the hyperparameters except the option of using the dynamic or fixed learning rate. We conduct the analysis using the CLUTRR-SG dataset since it is more complex and difficult (lower random performance).

## E    Experiments with Large Language Models

| Method | ProofWriter | | | ProofWriter$_{\text{distractor}}$ | | | CLUTRR-SG | | |
|---|---|---|---|---|---|---|---|---|---|
| | 2-h | 3-h | 5-h | 2-h | 3-h | 5-h | 2-h | 4-h | 6-h |
| GPT-3.5 $_{0-\text{shot}}$ | 58.4 | 56.4 | 53.7 | 49.1 | 47.1 | 45.3 | 35.6 | 16.0 | 18.5 |
| GPT-3.5 $_{8-\text{shot}}$ | 78.0 | 82.4 | 80.1 | 58.7 | 57.2 | 54.5 | 39.0 | 18.5 | 20.8 |
| RECKONING$_{\text{MT}}$ | **99.5** | **99.7** | **99.8** | **79.8** | **83.7** | **84.0** | **98.3** | **97.6** | **94.8** |

Table 10: Label accuracy of RECKONING on ProofWriter and CLUTRR-SG compared against a popular Large Language Model (LLM), GPT-3.5. We prompt GPT-3.5 in the zero-shot setting and also the 8-shot in-context learning setting. Models with MT are trained with the multi-task objective in the outer loop.

Recently, Large Language Models (LLMs) with large parameter sizes learned from human preferences have shown remarkable performance in language understanding and generation. These LLMs are powerful zero-shot and few-shot reasoners. Recent works find that LLMs learn to perform multi-step reasoning by first generating new reasoning chains and then predicting the answers. In this experiment, we benchmark the performance of a popular new LLM, GPT-3.5, on the two multi-hop reasoning datasets we used in our paper. We first evaluate GPT-3.5's zero-shot reasoning performance in predicting the correct answers. As Table 10 shows, zero-shot prompting GPT-3.5 significantly under-performs RECKONING's performance. GPT-3.5's performance improves on ProofWriter without distractors but still is behind the performance of RECKONING. When distractors are present in the context, RECKONING performs much better than zero-shot and few-shot GPT-3.5 prompting. This

highlights RECKONING's strength in disentangling irrelevant information from useful knowledge, an ability that even powerful LLMs like GPT-3.5 lack.

