# OpenReview forum: "RECKONING: Reasoning through Dynamic Knowledge Encoding"
_NeurIPS.cc/2023/Conference — NeurIPS 2023 poster_

### Official Review · Reviewer_5k3Z · 2023-07-05

**Soundness:** 2 fair
**Presentation:** 3 good
**Contribution:** 2 fair
**Rating:** 5
**Confidence:** 4

**Summary:**

The paper introduces a two-level learning algorithm called RECKONING, which enhances the in-context reasoning performance by addressing the issue of distractor facts. The algorithm consists of two learning steps, an inner and an outer loop, where the inner loop trains the model to encode contextual knowledge into its parameters through BP, and the outer loop teaches the model to use the updated parameters to answer questions. The experimental results on two multi-hop reasoning datasets indicate that RECKONING outperforms the ICR baseline by up to 4.5%. Additionally, RECKONING generalizes better to longer reasoning chains unseen during training, is more robust to distractors in the context.

**Strengths:**

Overall, the research topic presented in this work is of interest and importance, as it enhances transformer-based language models by incorporating in-context knowledge into parameters to improve reasoning ability. The proposed RECKONING consistently outperforms the conventional ICR-based methods. The proposed method's superiority is further confirmed through comprehensive experiments and ablation studies, which demonstrate its superiority in generalization to longer reasoning chains.

**Weaknesses:**

While the research topic is interesting and the proposed algorithm outperforms conventional in-context reasoning methods, there are some weaknesses in the paper. The proposed inner-loop of encoding knowledge into model parameters to improve reasoning abilities is limited in its incremental nature, which may lead to issues when facts from different contexts contradict each other. For example, the fact that "Peter is a son to Kyle" may be changed to "Kyle is a son to Peter" in another question's context. Additionally, the experiment part compares two baselines "No-Facts" and "Randome-Facts," which are too weak, and no other publicly available baselines have been compared.

**Questions:**

1. In Figure 3, it would be helpful to include more experimental results with training on both 4- and 6-hop questions, or both 2- and 4-hop questions to further validate the proposed algorithm's performance.
2. It would also be beneficial to include other subtasks like proof generation in the ProofWriter dataset to further explore the algorithm's capabilities.
3. Conducting more experiments on public pre-trained language models like LLaMA would provide additional validation for the proposed method's superiority.

**Limitations:**

I can not find any potential negative societal impact.

---

> ### Author Rebuttal · Authors · 2023-08-10
>
> ## Responses to Reviewer 5k3Z (R5)
>
> We thank the reviewer for their constructive comments, as well as describing our idea as interesting and important, and recognizing the benefits of our approach. We address the reviewer’s concerns and questions below:
>
>
>
> **W1: The reviewer argues that if there are contradictory facts between questions, it may lead to issues with the model performance because of the incremental nature of the inner-loop**
>
>
>
> We first clarify the inference process of RECKONING. For each new question, we start from the trained meta parameters of the model (which does not contain any background knowledge for the test set questions) and do a few steps of gradient updates to encode the background knowledge. Next, we directly evaluate the updated parameters on the question. After that, we discard this locally updated model. For the next questions, we start from the trained meta parameters again, which do not contain any information of the background knowledge from the previous questions. Thus, the inner-loop in our proposed approach is not incremental. The current question’s inner-loop does not depend on the previous question's inner-loop. Each question’s inner-loop is unique to the associated question during inference. Suppose the previous question’s context and the current question’s context contradict each other. In that case, the model will not be affected since the trained meta parameters will not contain information on the previous question’s context when we do inference on the current question.
>
>
>
> **W2: The reviewer argues that the experiments contain only weak baselines and no public baselines compared**
>
> We have added more baselines; see our response to Reviewer 1’s Q3. For a stronger baseline, we have finetuned GPT-XL-Lora with in-context reasoning. We show that on ProofWriter-5-hop, RECKONING (70.2) still improves over this baseline (65.0) when there are distractors in the context of a question. For stronger public baselines, in our supplementary material, we report the performance of GPT-3.5 (text-davinci-003) on ProofWriter and CLUTRR (Table 9, supplementary material). We include the results below as a reference:
>
>
> | | | ProofWriter |      |      |ProofWriter (distractor)|  | |CLUTRR |      |
> |-----------------|------|:-----------:|------|------|:-----------:|------|------|:-------:|------|
> | Method          | 2-h  | 3-h         | 5-h  | 2-h  | 3-h         | 5-h   | 2-h  |   4-h  | 6-h  |
> | GPT-3.5 (0−shot)| 58.4 | 56.4        | 53.7 | 49.1 | 47.1        | 45.3  | 35.6 |   16.0 | 18.5 |
> | GPT-3.5 (8−shot)| 78.0 | 82.4        | 80.1 | 58.7 | 57.2        | 54.5  | 39.0 |   18.5 | 20.8 |
> | RECKONING       | **99.5** | **99.7**        | **99.8** | **79.8** | **83.7**        | **84.0**  | **98.3** |   **97.6** | **94.8** |
> |
>
> The results show that even a best-performing model like text-davinci-003 still fails to perform well on the reasoning tasks and does not generalize well when the context includes distractors. Compared to a public best-performing model, we show that RECKONING significantly benefits reasoning tasks under a systematic complex setting. For other public baselines, we argue that our idea and the proposed algorithm are novel and have not been done before. Thus, we believe that other public baselines on the two datasets are also weak and do not fit in the scope of this study. Specifically, the public baselines do not evaluate models under a systematic generalization setting.
>
>
>
> **Q1: The reviewer suggests experiments that mix training data from different hops of CLUTRR for a longer reasoning chain**
>
> We trained the model using RECKONING on a mixture of 2-hop, 4-hop, and 6-hop data. We report our results in the pdf as Figure 1. We show that the performance of FT-ICR and RECKONING is roughly equivalent at low-hop reasoning. However, RECKONING shows greater improvement when generalizing to OOD hops, similar to Figure 3 in our paper.
>
>
>
> **Q2: The reviewer suggests expanding our proposed approach to the proof generation task**
>
> We thank the reviewer for this interesting idea! We note that our results in Table 4 show which facts the model recalls as relevant for reasoning about the question, which can be viewed as the first step of a proof, since the model identifies the relevant facts for reaching the correct answer. Extending RECKONING for the “full” proof generation task would require greater changes that we couldn’t implement in the rebuttal window. We will explore this and present the results in the camera-ready.
>
>
> **Q3: The reviewer suggests conducting more experiments on public pre-trained language models like LLaMA**
>
> This is a great suggestion! As our responses to W2 and Reviewer 2’s W1/Q1, we have demonstrated that large pre-trained language models like GPT-3.5 and ChatGPT still struggle with complex reasoning and do not generalize well with distractors (Table 9 in the supplementary material). We will conduct more experiments on open-source language models like LLaMA in our revision.

---

> > ### Author Response · Authors · 2023-08-19
> > **Thank you for improving your rating!**
> >
> > Many thanks to the reviewer for their helpful and constructive suggestions! We are grateful for the reviewer raising their score from 4 to 5.

---

### Official Review · Reviewer_W7M8 · 2023-07-06

**Soundness:** 3 good
**Presentation:** 4 excellent
**Contribution:** 2 fair
**Rating:** 6
**Confidence:** 4

**Summary:**

This work aims to solve reasoning tasks where models need to rely on knowledge provided as part of the task input. Motivated by the fact that pre-trained models encode a lot of irrelevant facts and they are good at retrieving the relevant ones for solving a downstream task, this work proposes an algorithm, RECKONING, that updates a model’s parameters to memorize the facts on the fly and then answer the question without having those facts as input explicitly. Experiments show that the proposed method can improve the model performance compared with the in-context reasoning baseline. The resulting model also generalizes better to examples requiring longer reasoning chains, is more robust to distractors and even more efficient in a certain setting.

**Strengths:**

-	Originality: Unlike prior works that propose different ways to filter irrelevant facts explicitly, this work makes use of the pre-trained LM to subsume all the facts and filter the irrelevant facts on its own. The idea is interesting and makes sense.
-	Quality: To validate the effectiveness, the work conducts sufficient experiments. The results demonstrate the superiority of the method in scenarios where a longer reasoning chain is needed or there exist a lot of distractors, which well motivate this study. An analysis is also provided to address the concern about the computation efficiency of the method.
-	Clarity: The algorithm is well explained despite requiring some effort to understand it.


**Weaknesses:**

-	In the standard setting where there are no added distractors, the improvement brought by the method is actually minimal (Table 1).
-	Also, the need to update the parameters on the fly makes the algorithm hard to be applied on large language models, which might have a greater capability in filtering irrelevant facts in-context already. I would suggest extending the idea to these large LMs by updating a subset of parameters (like adapters do).
-	One potential disadvantage of the method could be that the knowledge facts contain a lot of noise or toxic content that may undermine the model’s basic ability for reasoning. In this regard, methods that first filter irrelevant facts and then provide the remaining facts in-context do not have such an issue.


**Questions:**

1.	For the Random-Facts baseline, do you mean the random facts are provided during inference or also during training? Do you train it with the multi-task objective?

**Limitations:**

No limitation is discussed.

---

> ### Author Rebuttal · Authors · 2023-08-10
>
> ## Responses to Reviewer W7M8 (R4)
>
> We thank the reviewer for their constructive comments and viewing our idea as interesting and sound, our experiments as sufficient and well-motivated toward the study, and our algorithm as well explained. We address the reviewer’s comments and questions below:
>
>
> **W1: The reviewer argues that in the standard setting, where there are no added distractors, the improvement brought by the method is minimal (Table 1 of our paper)**
>
> In our paper, we mainly focus on the problem of generalization in more complex settings with distractors and longer reasoning chains. While our results in Table 1 show a small improvement for RECKONING, the baseline (FT-ICR) is also quite strong in this idealized setting, and our results are partly a sanity check that RECKONING works as well (even slightly better!). However, RECKONING exhibits even stronger improvements when idealized conditions are removed and the model has to generalize out of distribution and handle noisy inputs.
>
>
>
> **W2: The reviewer wonders if the performance gain of RECKONING will generalize to larger language models**
>
>
>
> This is an interesting question. First, we want to show that more advanced large language models like GPT-3.5 (text-davinci-003) still fail to generalize when there are distractors, i.e., irrelevant information present in the context. Below our evaluation results on ProofWriter using GPT-3.5:
>
> | | | ProofWriter |      |      |ProofWriter (distractor)|  | |CLUTRR |      |
> |-----------------|------|:-----------:|------|------|:-----------:|------|------|:-------:|------|
> | Method          | 2-h  | 3-h         | 5-h  | 2-h  | 3-h         | 5-h   | 2-h  |   4-h  | 6-h  |
> | GPT-3.5 (0−shot)| 58.4 | 56.4        | 53.7 | 49.1 | 47.1        | 45.3  | 35.6 |   16.0 | 18.5 |
> | GPT-3.5 (8−shot)| 78.0 | 82.4        | 80.1 | 58.7 | 57.2        | 54.5  | 39.0 |   18.5 | 20.8 |
> | RECKONING       | **99.5** | **99.7**        | **99.8** | **79.8** | **83.7**        | **84.0**  | **98.3** |   **97.6** | **94.8** |
> |
>
> While the comparison is not perfect since we cannot tune GPT3.5, we note that the performance drop in the distractor setting on ProofWriter is much greater than for RECKONING.
>
>
>
> As a closer comparison, when we apply RECKONING to larger language models, we see similar improvements. We applied RECKONING to GPT2-XL (1.5B) with LoRA and evaluated on ProofWriter 5-hop with all distractors. Compared to FT-ICR’s performance (**65.0**), RECKONING’s performance (**70.2**) is **5.2** percentage points higher, demonstrating that RECKONING’s benefits still appear with larger language models.
>
>
>
> **W3: The reviewer argues that noise or toxic content in knowledge may undermine the model’s basic ability for reasoning. They propose that methods that filter irrelevant facts ahead of time could be more effective than RECKONING**
>
> This is an interesting point. However, we note that RECKONING is not in contradiction with a method that first filters irrelevant facts before providing them to the model for reasoning. Facts could be filtered before being provided to RECKONING too. We believe that combining these two approaches might work complementarily since existing approaches to filter irrelevant facts are not perfect.
>
>
>
> **Q1: The reviewer asks if we provide random facts during inference or during training, and do we train the random facts baseline with the multi-task objective**
>
> We provide random facts both during training and during inference. We train the random facts baseline without the multi-task objective.

---

### Official Review · Reviewer_ZqBF · 2023-07-07

**Soundness:** 3 good
**Presentation:** 3 good
**Contribution:** 3 good
**Rating:** 6
**Confidence:** 4

**Summary:**

This paper introduces a novel method for addressing logical reasoning in natural language question answering, specifically when the required knowledge is part of the context. One major challenge highlighted in this paper is that the included knowledge often contains irrelevant information, which can mislead the reasoning process and negatively impact question-answering performance.
The proposed solution involves a bi-level optimization technique during inference. The technical key is utilizing the knowledge to quickly fine-tune the model at inference time, using a causal language modeling objective. This eliminates the need for the knowledge to be explicitly included in the context, as it is encoded within the model parameters. It also helps the model to focus on the relevant parts of the knowledge.
To evaluate the effectiveness of the approach, multiple baselines incorporating variations of in-context learning are compared. The experimental results demonstrate improved performance using the proposed method, particularly in terms of generalization over longer reasoning chains.


**Strengths:**

Pros:
The proposed approach is sound and interesting.  The paper is well-written.
They show that their inference-time optimization makes the model more robust to the redundant information.
It becomes more generalizable to longer chains of reasoning.
It is more efficient when multiple questions are asked based on the same context.


**Weaknesses:**

The run-time analysis is to some extent misleading. For the mentioned problem setting, in general, we assume each example comes with its own context that is the context is not often shared. Therefore, I guess we expect this model to be less efficient in general for reasoning as it needs the bi-level optimization at inference time. I think highlighting the efficiency in the multi-question setting looks a bit far fetch.


**Questions:**

—While it is expected that the model learns to focus on the relevant parts of the knowledge, it is not very clear to me why the approach should make the model more generalizable to longer chains of reasoning? Any intuition?

—Is there any SOTA results  better than what you report in this paper in your baseline variations?

-- As it is explained in the paper, for the multiple question setting some parts of the knowledge are relevant for one question while irrelevant for others. Since the model is tuned once and the same model is used for answering all questions then the performance should be lower in this case, right?

**Limitations:**

The limitation section is not included in the paper.

---

> ### Author Rebuttal · Authors · 2023-08-10
>
> ## Responses to Reviewer ZqBF (R3)
>
> Dear reviewer, thank you for the constructive comments and we appreciate your time and effort. We thank the reviewer for recognizing our approach as interesting and sound, and our paper as well-written. We are encouraged to see the reviewer acknowledging our contributions. We address the reviewer’s concerns and questions below:
>
> **W1: The reviewer argues that the runtime analysis in the multi-question setting does not demonstrate RECKONING’s efficiency because the context is not often shared in general**
>
> We argue that the setting where multiple questions share the same context is a common problem setting. For example, reading comprehension (e.g., SQuAD) typically involves reading a passage and answering multiple questions about it. In our own experiments, the ProofWriter typically contains fact sets about which multiple questions can be asked. Certain facts are distractors for particular questions and relevant for others. One of the benefits of RECKONING is that it can answer multiple questions based on the same set of facts, and our evaluation uses this property to improve the evaluation speed in multi-question settings.
>
>
>
> **Q1: The reviewer asks about the intuition that our method makes the model more generalizable to longer chains of reasoning**
>
> This is a good question. Our hypothesis is that RECKONING is more generalizable to longer reasoning chains because it encodes the multiple pieces of knowledge as separate sequences in a batch, which may lead to less of a length distribution shift, since the maximum sequence length is roughly equivalent for most facts. For in-context reasoning, reasoning is performed through the forward pass using the attention mechanism, which may be more likely to overfit to training length as the facts are concatenated into a single sequence.
>
> **Q2: The reviewer wonders if there are SOTA results on the baseline variations**
>
> We also included an advanced large language model, GPT-3.5 (text-davinci-003) as one of the baselines. We show that RECKONING outperforms GPT-3.5 both on ProofWriter and CLUTRR:
>
> |                 |      | ProofWriter |     |   |ProofWriter (distractor)| |  |CLUTRR |      |
> |-----------------|------|:-----------:|------|------|:-----------:|------|------|:-------:|------|
> | Method          | 2-h  | 3-h  | 5-h             | 2-h  | 3-h         | 5-h   | 2-h  |   4-h  | 6-h  |
> | GPT-3.5 (0−shot)| 58.4 | 56.4  | 53.7    | 49.1 | 47.1        | 45.3  | 35.6 |   16.0 | 18.5 |
> | GPT-3.5 (8−shot)| 78.0 | 82.4  | 80.1    | 58.7 | 57.2        | 54.5  | 39.0 |   18.5 | 20.8 |
> | RECKONING       | **99.5** | **99.7** | **99.8**       | **79.8** | **83.7**        | **84.0**  | **98.3** |   **97.6** | **94.8** |
> |
>
> We highlight the significant performance gain when there are distractors in the context of a question. While performance drops are observed for both approaches, the drop for GPT-3.5 is greater than for RECKONING.
>
> **Q3: Should the performance be lower when multiple questions share the same context and some parts of the knowledge are relevant for one question while irrelevant for others**
>
> Compared to the setting where each question is associated with a unique context with no irrelevant information, yes, the performance is lower when the setting changes to multiple questions sharing the same context. However, we show in our experiment (Figure 5, paper) that models trained with our algorithm are much more robust when there are distractors (irrelevant information) in the context, compared to the in-context reasoning baseline.

---

> > ### Comment · Reviewer_ZqBF · 2023-08-16
> >
> > I have read the author's response and other reviews and discussions. Thank the authors for further clarifications on my questions.  I was already positive about this paper. My score remains unchanged.

---

> > > ### Author Response · Authors · 2023-08-19
> > > **Thanks for your encouraging comments!**
> > >
> > > We are encouraged to see the reviewer being positive about our paper. We are grateful for the reviewer's constructive feedback!

---

### Official Review · Reviewer_UhBM · 2023-07-13

**Soundness:** 3 good
**Presentation:** 3 good
**Contribution:** 2 fair
**Rating:** 7
**Confidence:** 4

**Summary:**

Summary:
The paper introduces RECKONING, a bi-level learning algorithm designed to improve reasoning in transformer-based language models. RECKONING encodes contextual knowledge into the model's parameters using gradient updates, allowing the model to answer questions based on its updated parameters. The authors demonstrate, through experiments on two multi-hop reasoning datasets, that RECKONING outperforms an in-context reasoning baseline and is more robust to distractions, generalizes better to longer reasoning chains, and is more computationally efficient under certain conditions.

Contributions:
  1. Propose a bi-level learning algorithm, RECKONING, to teach language models to reason by updating their parametric knowledge through back-propagation.
  2. Conduct experiments on two multi-hop reasoning datasets, ProofWriter, and CLUTRR-SG, showing that RECKONING outperforms the in-context reasoning baseline and provides several other benefits.
  3. Provide analyses on the ability of RECKONING to memorize knowledge, measure its performance under distractor conditions, and analyze its run-time efficiency, etc.

**Strengths:**

  1. the idea of encoding knowledge into the model's parameters through gradient updates is an interesting and novel idea in the field of natural language reasoning
  2. RECKONING demonstrates better performance than the baseline model in several aspects, including better generalization to longer reasoning chains and its robustness to distractions
  3. The paper is well-presented

**Weaknesses:**

The main weakness I see for this paper is the scope of its conducted experiments:
  1. The experiments are conducted on two synthetic multi-hop reasoning datasets, ProofWriter and CLUTRR-SG. While these analyses provide valuable insights, further evaluation on a broader range of real-world datasets would strengthen the generalizability of RECKONING's performance.
  2. The experiments are conducted only using the base GPT-2 model, which is really behind the state of the art.  It's hard to tell if on the best performing models we can still see this impovement.


**Questions:**

1. How would RECKONING perform with more complex and diverse real-world reasoning tasks beyond the synthetic multi-hop reasoning datasets used in the experiments (ProofWriter and CLUTTR-SG)?  E.g., HotpotQA?
2. Is this improvement agnostic to model architectures?  How does RECKONING do on seq2seq models?
3. In the multi-task training setting, does the choice of weighting different terms affect the performance of RECKONING? (from the paper is 1:1?)

---

> ### Author Rebuttal · Authors · 2023-08-10
>
> ## Responses to Reviewer UhBM (R2)
>
> We thank the reviewer for their helpful comments and are encouraged that the reviewer agrees that our algorithm is “an interesting and novel idea in the field of natural language reasoning” and recognizes its benefits. We also thank the reviewer for complimenting our paper as “well-presented.” We address the reviewer’s concerns and questions below:
>
> **W1/Q1: The reviewer asks how RECKONING would perform with more complex and diverse real-world reasoning tasks outside of the scope of synthetic data**
>
> Taking the reviewer’s suggestion, we evaluated RECKONING on FOLIO, a reasoning benchmark with first-order logical reasoning problems written by expert annotators based on real-world knowledge. We use the validation set as an in-house test set since the true test set has not been publicly released. In the training process, we randomly split the training data into train and validation sets. We show the evaluation results below:
>
> | Model                             | Acc  |
> |-----------------------------------|------|
> | FT-ICR                            | 53.3 |
> | GPT-3.5 (text-davinci-003) 0-shot | 45.1 |
> | GPT-3.5 (text-davinci-003) 8-shot | 52.9 |
> | chatGPT (gpt-3.5-turbo) 0-shot    | 40.0 |
> | chatGPT (gpt-3.5-turbo) 8-shot    | 42.6 |
> | RECKONING                         | **54.9** |
> |
>
> Our experiments show that RECKONING still outperforms the FT-ICR baseline and improves the performance compared to best-performing large language models like GPT-3.5 and ChatGPT, showing that even in less synthetic settings, RECKONING can still outperform other approaches.
>
> **Q2: The reviewer asks if the improvements are agnostic to model architectures and wonders how RECKONING would perform with seq2seq models**
>
> This is a great suggestion! We extended RECKONING on a T5-small model and evaluated it on ProofWriter-5-hop with all the distractors. The FT-ICR baseline with T5-small achieves 69.2, and RECKONING achieves 70.8. We show that RECKONING still outperforms the baseline. We will conduct more extensive experiments on seq2seq models in our future revision.
>
> **Q3: The reviewer wonders if the choice of weighting on the two terms of the multi-task objective affect the performance of RECKONING**
>
> This is an interesting point! In principle, the weighting would have some impact on the performance. However, we find that a 1:1 ratio works well in our study, so we stick to this setting due to the limited computation budget.

---

> > ### Comment · Reviewer_UhBM · 2023-08-15
> > **Author response addressed my questions**
> >
> > Thanks for taking the time to run additional experiments, and the experiment on FOLIO looks great.

---

> > > ### Author Response · Authors · 2023-08-15
> > > **Thanks for your encouraging comments!**
> > >
> > > We thank the reviewer for their encouraging response and constructive suggestions! We are grateful for the reviewer raising their score from 5 to 7.

---

### Official Review · Reviewer_ymTg · 2023-07-28

**Soundness:** 3 good
**Presentation:** 3 good
**Contribution:** 3 good
**Rating:** 7
**Confidence:** 3

**Summary:**

The paper presents a two-phase learning algorithm that 1) encodes background knowledge in the parameters of an LM through fine-tuning (instead of providing it in context), and 2) learns how to use and reason the encoded knowledge for a given question (?).

The paper shows better performance than in-context learning, and better generalization to longer reasoning chains.

**Strengths:**

- The paper is easy to read for the most part.
- The generalization experiments results support Reckoning as a useful approach.

**Weaknesses:**

- Most of my concerns / questions have to do with Phase 2 of the model updates. More details given below in questions.

**Questions:**

- Does the order in which the questions are seen effect the results?
- Which task from the Proofwriter dataset was actually used in this paper?
- Is the outer-loop of updates actually needed? Finetuning on knowledge and then trying to answer the question directly (without any more parameter updates) is a natural baseline to compare against. I believe this is slightly different from the FT-ICR setup.
- In any case, the performance difference between FT-ICR and Reckoning is only somewhat apparent on the CLUTTRR dataset. Can you conjecture why that might be the case?
- I think the bigger performance gaps are noticeable in the OOD generalization setup. Can you provide an intuition / analysis of why generalization gets better?
- At multiple places, the paper says phase 1 is to memorize the knowledge and phase 2 is to quickly memorize the given knowledge and perform reasoning. e.g. lines 108 to 111. Can you clarify why one needs 2 steps to memorize the said knowledge. Is the knowledge being memorize different in these two phases? Moreover, it seems memorize and "learn" are being used interchangeably, but I believe they are different. Overall, phase 2 application and its impact are confusing.
- If the knowledge from a question is being encoded in the model, how does the model handle contradictory information? For example: it is possible "A is B's son" in instance 1, but "B is A's son" in another instance. Is there an assumption that this is not possible?

**Limitations:**

-

---

> ### Author Rebuttal · Authors · 2023-08-10
>
> ## Responses to Reviewer ymTg (R1)
>
> We thank the reviewer for viewing our proposed bi-level learning algorithm as interesting and novel and for recognizing its benefits and contribution. Below, we address the questions:
>
> **Q1: The reviewer asks if the order of the questions affect the results**
>
> No, the order of the questions in the dataset should not affect the results. We randomly shuffle the train and test data for each run. For each experiment, we conduct three runs using three different random seeds and report the average.
>
> **Q2: The reviewer wonders which task from ProofWriter was used in our paper**
>
> We follow the deductive reasoning task originally proposed in RuleTaker, a work that ProofWriter is built on. The task is most similar to Task 1, proof generation, but we omit the proof part and only generate the answer.
>
> **Q3: The reviewer asks if the outer-loop of updates is needed**
>
> We evaluated four additional baselines related to the outer-loop updates to confirm this:
>
> -   **FT-KG**: Here we check, without outer loop optimization, can finetuning on all facts allow the model to memorize them and perform well on the questions when no additional information is given.
>
> -   **FT-KG-ICR**: Here we check, without outer loop optimization, can finetuning on all facts allow the model to memorize them and perform well on the questions when the background knowledge is given at test time too.
>
> -   **RECKONING-no-outer**: We check if RECKONING with inner-loop only (i.e., a single-level optimization) can perform well when no facts are provided in the question. Note that for each question, we start from the trained model that is not yet finetuned on that question’s facts.
>
> -   **RECKONING-no-outer (zero-shot)**: We do not train the model but directly evaluate it on the questions by dynamically doing a few gradient steps to encode the facts. We check if the model already can do inference-time knowledge encoding through gradient descent dynamically for each question, without bi-level optimization.Note that for each question, we start from the initial model that is not yet finetuned on that question’s facts.
>
> We report their performances below (also see Table 1, rebuttal pdf):
>
> |                                | Clutrr-2-hop | Clutrr-4-hop | Clutrr-6-hop |
> |--------------------------------|--------------|--------------|--------------|
> | FT-KG                          | 5.1          | 4.6          | 5.8          |
> | FT-KG-ICR                      | 5.1          | 4.6          | 5.8          |
> | RECKONING-no-outer (zero-shot) | 7.9          | 8.1          | 9.9          |
> | RECKONING-no-outer             | 20.7         | 12.9         | 10.3         |
> | RECKONING                      | **98.3**         | **97.6**         | **94.8**         |
> |
>
> |                                | Proof-2-hop | Proof-3-hop | Proof-5-hop |
> |--------------------------------|-------------|-------------|-------------|
> | FT-KG                          | 31.2        | 33.8        | 33.3        |
> | FT-KG-ICR                      | 32.5        | 32.4        | 33.6        |
> | RECKONING-no-outer (zero-shot) | 33.3        | 29.7        | 33.0        |
> | RECKONING-no-outer             | 17.6        | 14.2        | 6.8         |
> | RECKONING                      | **99.5**        | **99.7**        | **99.8**        |
> |
>
> As our evaluation results show, the baselines FT-KG and FT-KG-ICR perform close or under random (33.3% for ProofWriter and 5% for CLUTRR). The baselines that remove the outer loop also perform poorly, far below RECKONING’s performance. We highlight the importance of outer-loop optimization, indicating that it's necessary for the model to learn to dynamically do few-step knowledge encoding that supports the reasoning performance.
>
> **Q4/Q5: The reviewer asks why the performance difference between FT-ICR and RECKONING is only apparent on CLUTRR, and why bigger performance gaps are noticeable in the OOD generalization setup**
>
> Our results in Table 1 show small improvement for RECKONING as the baseline (FT-ICR) is also quite strong in this idealized setting. Here, our results are partly a sanity check that RECKONING works as well under ideal conditions (even slightly better!). However, RECKONING exhibits stronger improvements when idealized conditions are removed and the model has to generalize out of distribution and handle noisy inputs.
>
> **Q6: The reviewer asks us to clarify knowledge memorization and learning and motivate the importance of outer loop optimization**
>
> In RECKONING, we do not first fine-tune the model on all facts in the dataset and then learn to “quickly memorize the given knowledge and perform reasoning,” as the reviewer suggested. Instead, RECKONING does inference-time training on the fly. We define knowledge memorization as doing a few gradient updates on the facts. In the inner-loop, the model encodes the knowledge through these gradient updates. In the outer-loop, the model uses the encoded knowledge to perform reasoning. Please see more context on the inference process in our general response (point 1). To teach models the ability to do this kind of inference-time learning, i.e., gradient-based knowledge encoding that supports reasoning, bi-level optimization is important. As we have shown in our response to Q3, without the outer loop, models perform poorly.
>
> **Q7: The reviewer asks how does the model handle contradictory facts across examples**
>
> RECKONING would be able to handle this case. As mentioned in the general response, at inference time, RECKONING falls back to learned “meta-parameters” after every processed example, wiping the slate clean for the next example. As a result, contradictory knowledge between examples do not contaminate each other. RECKONING would likely not work if there were contradictory facts for the **same** example, but to the best of our knowledge, even the best performing models cannot reliably handle contradictory information in the context well.

---

> > ### Comment · Reviewer_ymTg · 2023-08-18
> > **Rebuttal Response**
> >
> > Thank you for the response and for clarifying my questions. The new results are also interesting to learn about. I've also read the other reviews and responses. I was already positive about the paper and I am happy to raise it a bit in light of the strong new results that've been reported during the rebuttal phase. Thanks!

---

> > > ### Author Response · Authors · 2023-08-19
> > > **Thanks for your encouraging comments!**
> > >
> > > We appreciate the reviewer's helpful suggestions and encouraging response to our rebuttal! We are grateful for the reviewer raising their score from 6 to 7.

---

### Author Rebuttal · Authors · 2023-08-10


## General responses to all reviewers
We would like to thank the reviewers for providing us with thoughtful comments and constructive feedback. We appreciate that the reviewers recognize our proposed **bi-level learning algorithm** for language reasoning as novel/interesting (R2, R3, R4, R5), useful/important (R1, R5), sound (R3), and our work conducts sufficient **experiments** (R4). We are encouraged that the reviewers see our **paper** as well-written (R2, R3) and well-explained (R4).

We address the general concerns below:

**(1) A few reviewers asked us to clarify the inference process of RECKONING and how the knowledge-encoding and question-answering were conducted.**

Our proposed bi-level learning algorithm follows the idea of Model Agnostic Meta-Learning (MAML), where the objective is learning to do few-shot learning for downstream classification tasks. In the case of RECKONING, we are learning to do fast language modeling (e.g., knowledge encoding) for question answering. During inference time, we start from the trained meta parameters and train the model to do language modeling on the given facts using a few steps of gradient descent. Then we use the updated parameters to answer a question. After this, we discard the updated parameters and recover the trained meta parameters for the next question. This inference step is done for each new question. i.e., for each new question, we start from the same trained meta parameters. However, this ability to do inference-time gradient-based learning requires us to use a bi-level optimization algorithm during training to obtain the trained meta parameters.

**(2) Some of the reviewers ask how the performance of RECKONING would generalize with larger language models and real-world non-synthetic datasets.**

In response to this, we applied RECKONING to GPT2-XL (1.5B, 15 times bigger than the GPT2-small model used in our paper.) with LoRA. We evaluated on ProofWriter 5-hop with all distractors. Compared to FT-ICR’s performance (**65.0**), RECKONING’s performance (**70.2**) is **5.2** percentage points higher. We demonstrate that  RECKONING still shows significant benefits when scaling up to larger models.

To validate RECKONING’s performance on real-world reasoning tasks, we also evaluated on FOLIO, a complex logical reasoning benchmark involving real-world examples. We report the performances below:

| Model                             | Acc  |
|-----------------------------------|------|
| FT-ICR                            | 53.3 |
| GPT-3.5 (text-davinci-003) 0-shot | 45.1 |
| GPT-3.5 (text-davinci-003) 8-shot | 52.9 |
| chatGPT (gpt-3.5-turbo) 0-shot    | 40.0 |
| chatGPT (gpt-3.5-turbo) 8-shot    | 42.6 |
| RECKONING                         | **54.9** |
|

We can see that RECKONING still performs better than the in-context reasoning baseline, and it even surpasses best-performing large language models (LLMs) like GPT-3.5 and chatGPT.

In addition, we show that although the two datasets we used are synthetic, even best-performing models like GPT-3.5 still fail under the systematic generalization tests (Table 9 in the supplementary material):
|                 |      | ProofWriter |      |      |ProofWriter (distractor)|  | |CLUTRR |      |
|-----------------|------|:-----------:|------|------|:-----------:|------|------|:-------:|------|
| Method          | 2-h  | 3-h         | 5-h  | 2-h  | 3-h         | 5-h   | 2-h  |   4-h  | 6-h  |
| GPT-3.5 (0−shot)| 58.4 | 56.4        | 53.7 | 49.1 | 47.1        | 45.3  | 35.6 |   16.0 | 18.5 |
| GPT-3.5 (8−shot)| 78.0 | 82.4        | 80.1 | 58.7 | 57.2        | 54.5  | 39.0 |   18.5 | 20.8 |
| RECKONING       | **99.5** | **99.7**        | **99.8** | **79.8** | **83.7**        | **84.0**  | **98.3** |   **97.6** | **94.8** |
|

We can see GPT-3.5 especially struggles when there are distractors in the context of the questions. In general, we use synthetic datasets to allow us to control changes to the fact base (e.g., distractors and longer reasoning chains) such that we can systematically evaluate how sensitive in-context reasoning and RECKONING are to these factors. Our proposed learning algorithm improves model performance in more complex and challenging settings where factors like distractors and longer reasoning chains can be systematically evaluated.

---

### Decision · Program_Chairs · 2023-09-21

**Decision:**

Accept (poster)

**Comment:**

This paper proposes to encode knowledge through gradient updates instead of through concatenation of context and shows this leads to better robustness and generalization. Overall reviewers agreed the idea is novel, the paper well-written, and results show convincingly better generalization to longer reasoning chains.